# Methylphosphonate Degradation and Salt-Tolerance Genes of Two Novel Halophilic *Marivita* Metagenome-Assembled Genomes from Unrestored Solar Salterns

**DOI:** 10.3390/genes13010148

**Published:** 2022-01-15

**Authors:** Clifton P. Bueno de Mesquita, Jinglie Zhou, Susanna Theroux, Susannah G. Tringe

**Affiliations:** 1Department of Energy Joint Genome Institute, Lawrence Berkeley National Laboratory, Berkeley, CA 94720, USA; cliff.buenodemesquita@lbl.gov (C.P.B.d.M.); jingliezhou@gmail.com (J.Z.); 2Southern California Coastal Water Research Project, Costa Mesa, CA 92626, USA; susannat@sccwrp.org; 3Environmental Genomics and Systems Biology Division, Lawrence Berkeley National Laboratory, Berkeley, CA 94720, USA

**Keywords:** methanogenesis, methylphosphonate, salt tolerance, marine bacteria, phylogenomics, hypersaline

## Abstract

Aerobic bacteria that degrade methylphosphonates and produce methane as a byproduct have emerged as key players in marine carbon and phosphorus cycles. Here, we present two new draft genome sequences of the genus *Marivita* that were assembled from metagenomes from hypersaline former industrial salterns and compare them to five other *Marivita* reference genomes. Phylogenetic analyses suggest that both of these metagenome-assembled genomes (MAGs) represent new species in the genus. Average nucleotide identities to the closest taxon were <85%. The MAGs were assembled with SPAdes, binned with MetaBAT, and curated with scaffold extension and reassembly. Both genomes contained the *phnCDEGHIJLMP* suite of genes encoding the full C-P lyase pathway of methylphosphonate degradation and were significantly more abundant in two former industrial salterns than in nearby reference and restored wetlands, which have lower salinity levels and lower methane emissions than the salterns. These organisms contain a variety of compatible solute biosynthesis and transporter genes to cope with high salinity levels but harbor only slightly acidic proteomes (mean isoelectric point of 6.48).

## 1. Introduction

Biogenic methane production in nature has historically been attributed to anaerobic decomposition of organic matter performed by archaea. While this is the dominant known source of naturally produced methane, recent work has elucidated additional pathways performed by members of the bacterial domain in aerobic environments. This body of work has its origins in observations of elevated methane emissions from aerobic marine environments, a phenomenon which became known as the “methane paradox” [1]. Subsequent work attempting to explain the methane paradox in both oceans and freshwater ecosystems over the past 15 years has elucidated two new pathways of aerobic methane production by bacteria—methylphosphonate degradation via the now well-described C-P lyase pathway [1,2,3,4,5], and methylamine degradation via a yet-undescribed biochemical pathway involving aspartate aminotransferase [6].

The more well-known and better-described C-P lyase pathway of methylphosphonate degradation involves the suite of genes *phnCDE* to first import methylphosphonate into the cytoplasm and then *phnGHIJLM* to catalyze a sequence of four reactions [4,5,7]. Methane is produced as a byproduct in the third reaction, in which PhnJ catalyzes the conversion of α-D-ribose-1-methylphosphonate 5-phosphate to 5-phospho-α-D-ribose 1,2-cyclic phosphate [4,5,8]. Importantly, methylphosphonates can be produced by abundant marine organisms, including *Nitrosopumilus maritimus* [9] and *Pelagibacter ubique* [10]. Methylphosphonates are known to be used by microbes, both in marine and soil environments, as a source of phosphorus when other forms of phosphorus such as inorganic phosphate are limited [11,12,13].

Many genera of archaea and bacteria contain the *phnJ* gene, from Euryarchaeota to Proteobacteria and Cyanobacteria [14,15]. At the time of writing, *phnJ* genes were annotated in 15,935 genomes on the Integrated Microbial Genomes and Microbiomes (IMG/M) database [16]; of these, 15,875 genomes are identified to genus level and span 18 archaeal genera and 730 bacterial genera. *Marivita* (Rhodobacteraceae) is one of the bacterial genera harboring these genes that has been described as abundant in marine environments and important for ocean biogeochemical cycles [17]. *Marivita* have been isolated from different saline habitats around the globe, including the Korean coast [17,18,19,20], the Chesapeake Bay estuary [21], the saline Tuosu Lake in China [22], and from the marine dinoflagellate *Alexandrium catenella* [23]. They have been described as having optimal growth at 2–4% sodium chloride (NaCl) but are capable of tolerating salinities up to 10% NaCl. The two new metagenome-assembled genomes (MAGs) presented here come from sediment samples with 147.6 ppt (14.76%) and 166.8 ppt (16.68%) salinity (Cl^−^), which expands the known salinity tolerance of the genus.

Here, we describe two new draft genomes of two novel *Marivita* species, given the names *Marivita* sp. SBSPR1 and *Marivita* sp. SBSPR2. Since these metagenome-assembled genomes were assembled from hypersaline former industrial solar salterns with elevated methane emissions, we focused on describing methylphosphonate cycling genes and halotolerance genes (i.e., genes for compatible solute synthesis) present in the two genomes. We also conducted a phylogenetic analysis and comparison of shared and unique orthologous gene groups of the two MAGs and the five *Marivita* sister taxa with currently and publicly available *Marivita* genomes.

## 2. Materials and Methods

*Marivita* sp. SBSPR1 (named after South Bay Salt Pond R1) was assembled from metagenomic DNA sequence from a 5–15 cm deep sediment sample R1_B_D2 (IMG/M ID 3300007712, National Center for Biotechnology Information Sequence Read Archive (NCBI SRA) accession SRP098112) from an unrestored, hypersaline former industrial solar saltern as part of the study published by Zhou et al., 2021 [15]. *Marivita* sp. SBSPR2 (named after South Bay Salt Pond R2) was assembled from metagenomic DNA sequence from a 5–15 cm deep sediment sample R2_B_D2 (IMG/M ID 3300009061, NCBI SRA accession SRP118410), a 5–15 cm deep section of a sediment core from an adjacent unrestored hypersaline former industrial solar saltern. Industrial salt production occurred in these ponds from the 1850s to 2003. The two samples are characterized by high salinity (147.6 and 166.8 total ppt Cl^−^ for R1 and R2 samples, respectively), moderate temperature (25.0 and 26.2 °C), hypoxic but not completely anaerobic conditions (1.35 and 3.08 mg L^−1^ DO), and near neutral pH (7.36 and 7.80) [15]. Elevated methane (CH_4_) emissions were measured from both cores in the field (R1 = 1347, R2 = 1607 µmol CH_4_ m^−2^ d^−1^) [15]. The metagenomic sequencing, processing, and binning has been described previously [15,24]. For these two draft genomes, we began with MAGs constructed with MetaBAT [25], with a completeness of 99.05% and 96.29% and contamination of 0.74% and 3.17%, for *M.* sp. SBSPR1 and *M.* sp. SBSPR2, respectively, as calculated by CheckM version 1.0.18 [26], which would be considered “high-quality” by accepted standards [27]. The original MAGs were initially classified to genus level as *Marivita* by both Bin Annotation Tool [28] and the GTDB-Tk classifier version 1.1.0 [29].

We estimated the abundance of the two MAGs across the 24 samples collected by Zhou et al. (2021) using average read depth and then transformed this to counts per million of assembled reads [15]. The effect of site (4 separate wetlands) on MAG abundance was tested with a Kruskal–Wallis and Nemenyi post hoc tests, as assumptions for ANOVA and Tukey HSD were not met (Levene Test, *p* < 0.05, Shapiro–Wilk test, *p* < 0.05).

We used Geneious version 2021.1.1 [30] (Biomatters Ltd., Auckland, New Zealand) to remap paired-end reads to each scaffold and extend using reads partially overlapping the ends of the scaffolds as described elsewhere [24,31]. Performing two rounds of these steps improved the completeness of *M.* sp. SBSPR1 to 99.10% and *M.* sp. SBSPR2 to 97.34% and decreased the contamination of *M.* sp. SBSPR2 to 2.37%. Neither genome contained a full-length 16S rRNA gene, which can be difficult to recover during MAG assembly. The updated MAGs were again classified as *Marivita* by both Bin Annotation Tool and GTDB-Tk. The genomes were then uploaded to the Joint Genome Institute’s Integrated Microbial Genomes and Microbiomes database (IMG/M) [16] to undergo their comprehensive annotation procedure and are publicly available with IMG/M IDs 2930928012 and 2930931642.

To compare the two new genomes with their sister taxa in the *Marivita* genus, we used KBase [32] to download five reference genomes publicly available from the RefSeq [33] database. These reference genomes are *M. hallyeonensis* (NZ_FQXC01000020) [18,34], *M. geojedonensis* (NZ_PVTN01000001) [19,34], *M. lacus* (NZ_BMFC01000001) [22,35], *M. cryptomonadis* MP20-4 (NZ_JAFBXM010000001) [36], and *M. cryptomonadis* LZ-15-2 (NZ_SWKO01000011) [23]. For a broader phylogenetic perspective, the “Insert Set of Genomes Into SpeciesTree” tool (version 2.2.0) in Kbase [32] was used to place the seven *Marivita* genomes into a phylogenetic tree with 50 other genomes from RefSeq using a concatenated alignment of 49 single-copy clusters of orthologous groups (COGs). A more detailed phylogenetic tree of just the seven *Marivita* genomes was constructed by first identifying and aligning 120 universal single copy bacterial marker genes with GTDB-Tk and then building a consensus tree from the concatenated alignment with RAxML version 8.12 [37] with the PROTGAMMALG model of amino acid substitution, 1000 bootstraps, and *Bacillus subtilis* strain 168 (RefSeq GCF_000009045.1) as an outgroup. ProtTest3 version 3.4.2 [38] was used to select the best model of amino acid substitution (LG). Lastly, average nucleotide identity (ANI) was calculated with FastANI version 0.1.3 implemented in KBase. The KBase narrative for this project is publicly available under narrative ID 92727.

Prodigal [39] was used to predict protein-coding genes from the nucleotide sequences. We calculated mean isoelectric points for the seven proteomes with EMBOSS [40] using the “iep” tool on the Galaxy website platform (https://usegalaxy.org/ (accessed on 13 August 2021) [41] with a pH step of 0.5. Both proteinortho [42] and KEGG orthology (KO) [43] were used to analyze orthologous gene groups, and intersections among the genomes were plotted with *ComplexUpset* in R [44]. KO profiles were created for each genome with BLAST Koala [45]. Pfam [46] was used to classify genes from proteinortho. KOs involved in methylphosphonate cycling and salt tolerance were extracted from the KO tables. A list of KOs involved in methylphosphonate cycling was created based on the literature [4,5] and the BioCyc [47] and KEGG [43] databases. A list of KOs involved in salt tolerance was developed based on discussion of salt-tolerance-related genes and processes in the literature [48,49,50,51,52,53], and it includes KOs that are involved in compatible solute biosynthesis and transport (e.g., betaine and trehalose) and cation transporters. KO presence/absence was plotted as heatmaps with *pheatmap* in R [54].

For a broader comparison of *Marivita* and *phnJ* abundances in the salterns to a wide variety of saline and hypersaline ecosystems, we downloaded available metagenomic datasets on IMG/M from sediment samples of marine/coastal habitats with at least two samples (*n* = 613) and at least marine salinity (e.g., lagoon, ocean, estuary, salt marsh), as well as inland saline habitats such as hypersaline lakes and hot springs with at least two samples. *phnJ* counts were acquired from assembled metagenomes using the “Statistical Analysis” tool on IMG/M set to analyze KO counts per genome. Counts were then normalized with *DESeq2* [55] in R, and then the *phnJ* gene (K06163) was extracted from the full table. *Marivita* abundances were acquired by downloading the genus-level taxonomic tables with the “Statistical Analysis” tool and “Gene Count” measurement (not using coverage as it was not available for all metagenomes) on IMG/M and then calculating counts per million assembled metagenomic reads with the “Genome Size * Assembled” metadata column associated with each metagenome on IMG/M. Only metagenomes with at least 1000 genus-level classified reads prior to transformation were included (*n* = 590). Information about the metagenomes in this analysis can be found in Appendix A. All R analyses were performed with version 4.0.2 [56].

## 3. Results

According to the phylogenetic tree constructed with 49 single copy COGs, the MAGs were placed within the *Marivita* genus (Appendix A), in agreement with MAG classification tools such as Bin Annotation Tool and GTDB-Tk. In the more detailed tree of just *Marivita* reference genomes and the two MAGs using the set of 120 universal bacterial single-copy protein coding genes in GTDB-Tk, both genomes were placed taxonomically with the *Marivita* genus, yet appear to be unique species in the genus (Figure 1). This result is supported by ANI analysis, with ANI values among the MAGs and reference genomes ranging from 78 to 84, which suggests new species in the *Marivita* genus (Appendix A) [57,58,59]. Both genomes were most closely related to *M. lacus*. *M*. sp. SBSPR1 has the largest genome of the seven *Marivita* genomes presented here, while *M*. sp. SBSPR2 has the smallest. The MAGs have two of the three highest GC contents among the seven genomes (Figure 1).

Across all seven genomes, proteinortho identified 5367 orthologous gene groups, while KEGG annotation identified 2312 KOs. Among the seven *Marivita* genomes included in this analysis, according to proteinortho, there were 2305 orthologous genes shared among all seven taxa; there were 279 genes missing from the *Marivita* sp. SBSPR1 genome, 64 genes missing from *Marivita* sp. SBSPR2, and 74 additional genes missing from both (Figure 2a). These missing genes encompass a wide variety of functions, including various transporters (ABC, periplasmic, binding-protein-dependent systems) and various enzymes (e.g., citrate synthase, Acyl-CoA dehydrogenase, dimethylsulfoniopropionate lyase) (Appendix A). In terms of KOs, there were 91 KOs present in the other six genomes but not that of *M*. sp. SBSPR1, and likewise, 21 KOs were missing from *M*. sp. SBSPR2, as well as an additional 27 KOs missing from both genomes (Figure 2b). These included, for example, nitrate/nitrite transport system genes *nrtABC*, glucose/mannose transport system genes *gtsABC*, and N-methylhydantoinase *hyuAB* genes (Appendix A). There were also 12 KOs present in both MAGs and missing from the other five reference genomes. Half of these were D-xylose (*xylFGH*) and putative multiple sugar transport system genes *chvE* and *gguAB* (Appendix A). While genes unique to one genome were not calculated by proteinortho (by definition, as they are not shared), we identified 92 KOs unique to *M.* sp. SBSPR1 and 44 KOs unique to *M.* sp. SBSPR2 (Figure 2b, Appendix A).

With respect to methylphosphonate cycling genes, none of the genomes examined here contained the *mpnS* gene for methylphosphonate biosynthesis, but all contained the three methylphosphonate transporter genes *phnCDE* (Figure 3). Five of the seven genomes, including both MAGs, contained the full suite of genes *phnGHILMJ* for the C-P lyase methylphosphonate degradation pathway. It is also notable that the two unrestored salterns sampled here had among the highest *phnJ* gene counts of any saline or hypersaline sediment metagenomes on IMG/M (Figure 4a). Following production of methane during the formation of 5-phospho-a-ribosyl-1,2-cyclic phosphate catalyzed by *phnJ*, the degradation can proceed to produce either a-D-ribose 1,5-biphosphate or D-ribofuranose 5-phosphate. All seven genomes contained *phnP* to produce the former, while they all lacked *phnPP* to produce the latter. Alternatively, there is another described pathway of methylphosphonate degradation that does not involve C-P lyases [60]. However, none of the seven genomes contained *phnY* or *phnZ1*, the two genes involved in this non C-P lyase pathway.

The two MAGs were more abundant in two unrestored former salterns compared to a reference wetland and a restored saltern (Figure 5a). Furthermore, generally *Marivita* counts per million were greater in the two hypersaline former salterns sampled here than in other saline to hypersaline sediments such as lagoons, mangroves, open ocean sediments, or hypersaline lakes (Figure 4b), although some specific salt marshes and marine samples had greater counts. Greater abundances of the MAGs were associated with the greater methane emissions in the salterns, but these relationships were not linear (Figure 5b). MAG abundances followed a bell-shaped curve in relation to salinity, close to zero at near ocean salinity, peaking between 100 and 150 ppt Cl^−^, and declining in the most hypersaline saltern sediments with 200+ ppt Cl^−^ (Figure 5c).

The two MAGs and the other five reference genomes contained genes for the synthesis and transport of a variety of compatible solutes including betaine, ectoine, glutamine, glutamate, hydroxyectoine, proline, and trehalose, as well as potassium uptake uniporters (Figure 6). There were some differences in the presence/absence of these genes between the two MAGs and between the MAGs and the other reference genomes. For example, *M*. sp. SBSPR1 lacked genes for trehalose synthesis and transport but was the only genome containing the *sdmt* gene for betaine biosynthesis from sarcosine. On the other hand, *M*. sp. SBSPR2 lacked trehalose synthesis genes but was the only genome containing trehalose transporter genes. *M*. sp. SBSPR2 was also the only genome containing *Vnx1*, a sodium and hydrogen antiporter for sodium extrusion. *M.* sp. SBSPR2 and *M.* lacus were the only two genomes with the *ectD* gene for hydroxyectoine synthesis. The proteomes were only slightly acidic, with mean isoelectric points ranging from pH 6.33 to 6.51 (Figure 1), which is greater than the numbers reported for specialized halophilic archaea using the “salt in” strategy [61]. However, the genomes do show asymmetrical bimodal isoelectric point profiles typical of halophilic organisms with a large peak around 5 and a smaller peak around 10 (Appendix A).

## 4. Discussion

*Marivita* taxa have been isolated from diverse marine environments and as part of the *Roseobacter* clade are important players in marine biogeochemical cycles [17,62,63]. Indeed, in a broad survey of other saline metagenomes on IMG/M, *Marivita* were present in most sample types except for lagoon sediments and inland saline hot spring sediments, the latter of which is not surprising due to the extreme temperatures. Previous laboratory growth experiments on *Marivita* cultures demonstrated a salt tolerance up to at least 10% in two taxa [22] but optimal growth at 2–4% salinity and maximum tolerances between 5 and 9% for the majority of the described *Marivita* species [18,19,20,21,22,23]. It is thus notable that we found two *Marivita* MAGs in samples with up to 23% salinity, which expands the known salt tolerance of the genus.

All seven *Marivita* genomes contained genes for the synthesis and transport of multiple compatible solutes, including betaine, ectoine, glutamine, glutamate, and proline, and some genomes additionally contained genes for trehalose and hydroxyectoine. These highly soluble, low-molecular-weight compounds are used by bacteria to maintain osmotic equilibrium with the environment without increasing intracellular salt concentrations [64]. The *Marivita* proteomes were slightly acidic on average, similar to other aerobic marine bacteria such as *Aliivibrio fischeri* (6.52) and *Alteromonas macleodii* (6.46) [61], with asymmetric bimodal isoelectric profiles (Appendix A). Their average isoelectric points (p*I*s) of 6.48 are considerably higher than extremely halophilic organisms such as the archaeon *Halobacterium* NRC-1 (5.03) and the bacterium *Salinibacter ruber* (5.92), both of which require high salt concentrations and utilize the “salt in” strategy of increasing cytoplasm salt concentrations with potassium [61]. Furthermore, 6.48 is also higher than the p*I* of a *Methanosarcinaceae* sp. genome isolated from the same Pond R1, which had a mean p*I* of 5.90 [24], similar to *Salinibacter ruber*. As with p*I*, GC contents of the *Marivita* MAGs (~62%) were also similar to the other *Marivita* genomes and are in the range of other reported halophilic organisms [65]. Thus, neither of these MAGs possess salinity adaptation mechanisms clearly distinct from those found in other *Marivita* species. A broad analysis of gene content did not suggest any additional adaptation mechanisms, although numerous differences in transporter genes suggest these may help these organisms survive in this unique biogeochemical environment.

All of these data together suggest that despite being found in 15–23% salinities, these two *Marivita* MAGs are capable of growing in a wide range of salinities and are not specialized to only extremely hypersaline environments, but this remains to be confirmed by laboratory growth experiments. We plan to conduct follow up studies on the results presented in this paper with efforts to culture these organisms and run growth experiments with methylphosphonate substrates, different levels of salinity, and other standard variables.

Given that the hypersaline unrestored salterns studied here had elevated methane emissions [15], we were particularly interested in the capability of *Marivita* to produce methane from methylphosphonate degradation. Indeed, both MAGs as well as genomes of three other taxa (*M. lacus*, *M. cryptomonadis* MP20-4, and *M. cryptomonadis* LZ-15-2) contained genes to import methylphosphonate into the cytosol (*phnCDE*) and degrade it to a-D-ribose 1,5-biphosphate via the C-P lyase pathway (*phnGHILMJP*). From the genomic data alone, *M. hallyeonensis* and *M. geojedonensis* do not appear to be capable of degrading methylphosphonate.

The *phnJ* gene involved in the methane production step of methylphosphonate degradation was notably high in unrestored salterns compared to all of the other ecosystem types, which can be attributed to the higher abundances of *phnJ*-containing taxa such as the two *Marivita* genomes presented here as well as other taxa such as *Roseobacter* [15], likely as a result of phosphate-limited conditions. *phnJ* abundances have been found to be negatively correlated with inorganic phosphate concentrations in both ocean water [11] and soils [12]. In our data, this trend is also true; *phnJ* abundance increases with increasing phosphorus limitation as measured by both decreasing inorganic phosphate levels and increasing inorganic N:P ratios (Appendix A). The unrestored salterns are phosphorus-limited relative to the reference wetland where both total N:total P ratios and inorganic N:inorganic P ratios are much lower than in the unrestored ponds [15] (Appendix A). In such phosphate-limited conditions, methylphosphonates can provide an alternative source of phosphorus for *Marivita*, as has been shown for several other taxa such as the cyanobacteria *Nodularia spumigena* [14] and *Trichodesmium* [66], the Alphaproteobacteria *Pelagibacterales* [13], *Agrobacterium*, *Rhizobium* [5], and the Gammaproteobacteria *Pantoea* [5].

Neither of the two *Marivita* MAGs nor other *Marivita* reference genomes contained *mpnS* to synthesize methylphosphonate. It has been previously shown that other abundant marine taxa such as the ammonia-oxidizing Thaumarchaeota taxa *Nitrosopumilus maritimus* and *Pelagibacter ubique* can synthesize methylphosphonate, so these or other marine organisms could be a source of methylphosphonates in this system [9,10]. While *Nitrosopumilus* was not abundant in the unrestored salterns [15], it was abundant in the reference wetland and in San Francisco Bay area wetlands in general, and thus there could be both historic stores of methylphosphonates from before the ponds became hypersaline, or inputs from surrounding areas. In fact, the *mpnS* gene (K18049) was completely absent in saltern R1 metagenomes and nearly absent (total counts of 0–6) in saltern R2 metagenomes [15], so it is unclear if there is currently any methylphosphonate production in the unrestored ponds themselves or if the only current inputs are influxes from the surrounding environments.

## 5. Conclusions

Here, we have presented the draft genome sequences of two new *Marivita* MAGs, designated *Marivita* sp. SBSPR1 and *Marivita* sp. SBSPR2, and compared them to their sister taxa in the *Marivita* genus. Both of these genomes likely represent new species in the *Marivita* genus; while clearly placed in *Marivita* in a reference genome tree, ANI values were less than 85%. *Marivita* sp. SBSPR1 and *Marivita* sp. SBSPR2 both contain genetic architecture for compatible solute transport and biosynthesis, likely enabling them to grow in the hypersaline samples from which the metagenomes were sourced. *Marivita* are ubiquitous marine taxa, which may play important biogeochemical roles including methylphosphonate degradation, not only in the open ocean but also in human-altered coastal wetlands.

## Figures and Tables

**Figure 1 genes-13-00148-f001:**
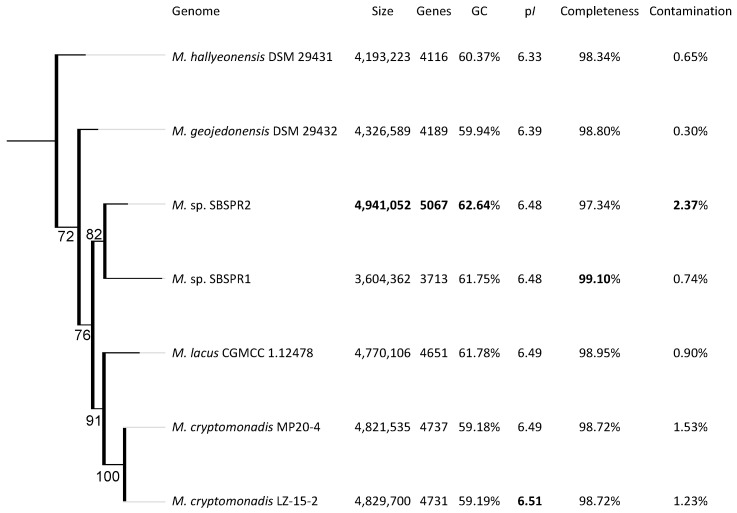
RAxML phylogenetic tree of the genus *Marivita*. The tree was built with a concatenated alignment of 120 single-copy bacterial genes and the PROTGAMMALG model of amino acid substitution. Branch labels show the bootstrap support, calculated with 1000 bootstraps. The genome size, number of protein coding genes, percentage G + C content, percentage completeness and contamination estimates from CheckM, and RefSeq ID are also shown. Bolded values highlight the greatest values in each column. The tree is rooted with *Bacillus subtilis* strain 168 as an outgroup (not shown).

**Figure 2 genes-13-00148-f002:**
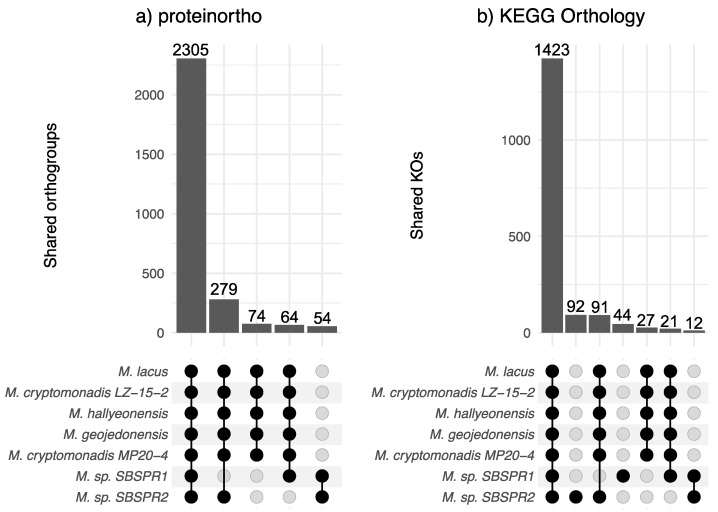
Shared orthologous gene groups in the genus *Marivita*. Intersections were calculated with (**a**) proteinortho and (**b**) KEGG orthology profiles. Displayed are intersections among all seven genomes, six other genomes but not *Marivita* sp. SBSPR1, six other genomes but not *Marivita* sp. SBSPR2, genes unique to both *M*. sp. SBSPR1 and SBSPR2, or genes present in only *M*. sp. SBSPR1 or *M*. sp. SBSPR2 (only applicable for shared KOs).

**Figure 3 genes-13-00148-f003:**
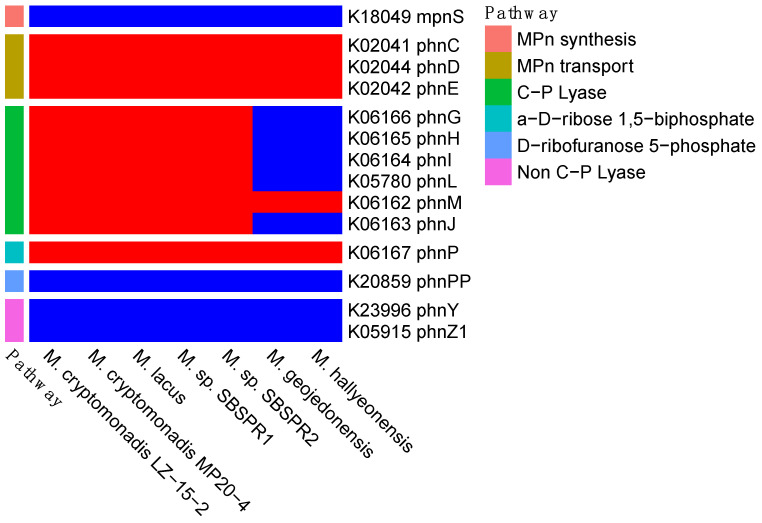
Presence (**red**) or absence (**blue**) methylphosphonate synthesis, transport, and degradation genes, with their abbreviated names and KEGG K number assignments. Shown here are the core C-P lyase pathway genes followed by two different post-methane pathways (*phnP* or *phnPP*), as well as the non-C-P lyase pathway.

**Figure 4 genes-13-00148-f004:**
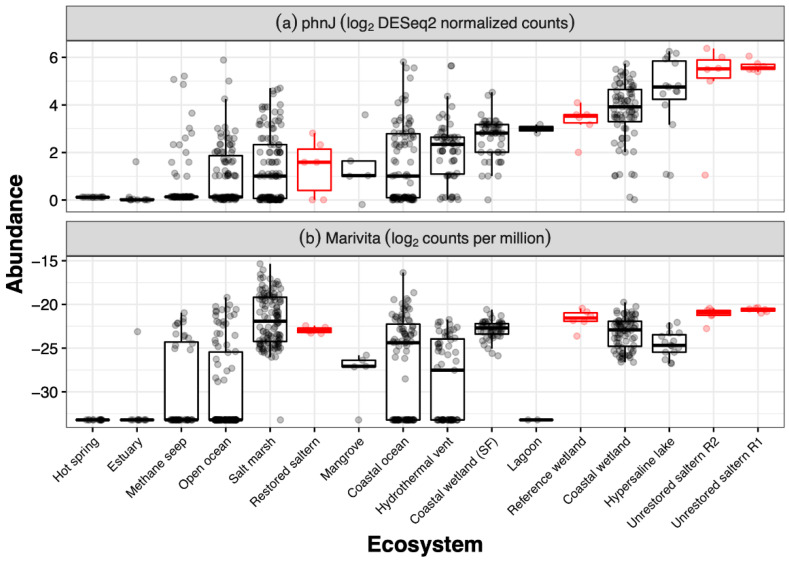
Abundance of (**a**) *phnJ* gene (K06163) and (**b**) *Marivita* from metagenomic sediment samples from various saline to hypersaline environments. Abundances are gene counts that were extracted from publicly available metagenomes on IMG/M using the “Statistical Analysis” tool and selecting feature type “KO” for (**a**) and “Genus” for (**b**) (see methods). Highlighted in red are the restored saltern, reference wetland, and the two unrestored salterns (R2, R1) described in Zhou et al. (2021). Note that open ocean sediments are separated from coastal ocean sediments. Furthermore, coastal wetlands from the San Francisco Bay and Delta (“Coastal wetland (SF)”) are separated from other coastal wetland samples due to their proximity to our sampling sites.

**Figure 5 genes-13-00148-f005:**
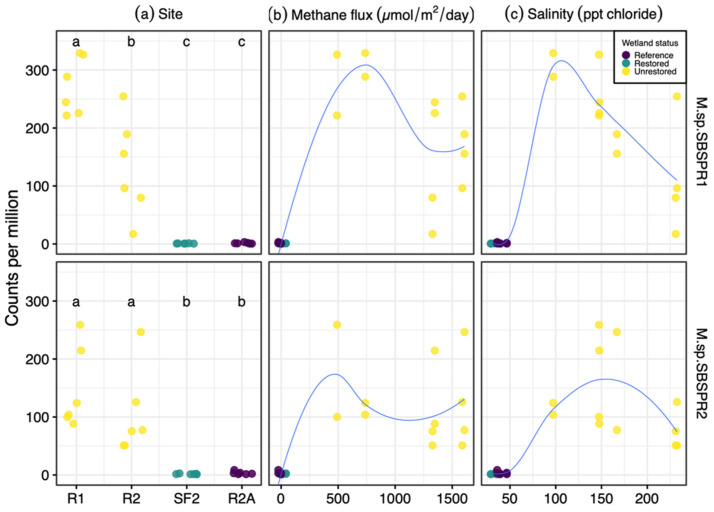
Abundance of *Marivita* sp. SBSPR1 and SBSPR2, expressed as counts per million assembled reads, across 24 metagenomic samples from Zhou et al. (2021) [15], organized by (**a**) site, (**b**) methane flux, and (**c**) salinity. Lines from loess functions are shown in (**b**,**c**). R1, R2, SF2, and R2A refer to site names as in Zhou et al. (2021). Different letters in (**a**) denote statistically significant pairwise differences (Nemenyi posthoc test, *p* < 0.05).

**Figure 6 genes-13-00148-f006:**
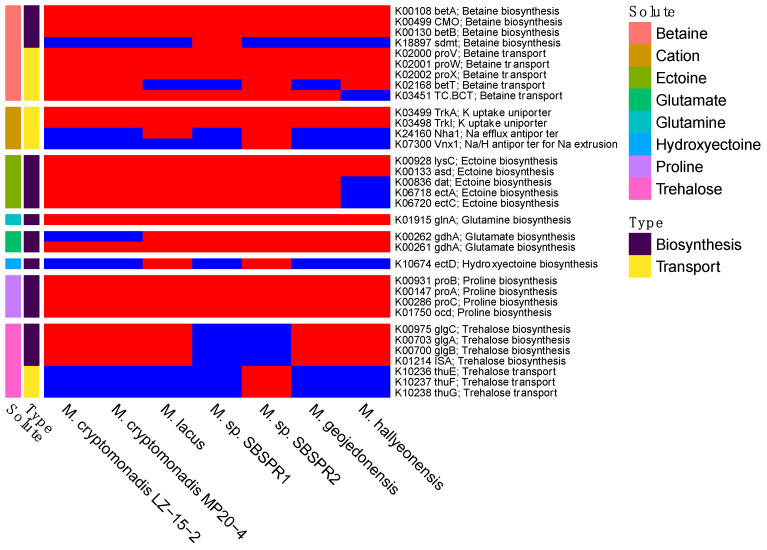
Presence (**red**) or absence (**blue**) of compatible solute or salt biosynthesis and transport genes, with their abbreviated names and KEGG K number assignments. Compounds include betaine, cations (K^+^, Na^+^, H^+^), ectoine, glutamine, glutamate, hydroxyectoine, proline, and trehalose.

## Data Availability

The two new genomes are publicly available at NCBI (JAIPUS000000000 and JAIPUT000000000) and IMG/M (2930928012 and 2930931642), with detailed metadata provided in GOLD (Ga0500780 and Ga0500781) for *M*. sp. SBSPR1 and *M.* sp. SBSPR2, respectively. The metagenomes from which they were derived are also available on IMG/M (3300026165 and 3300026251; GOLD IDs Gp0125921 and Gp0125927) and NCBI (SAMN06264691 and SAMN07631159) for *M*. sp. SBSPR1 and *M.* sp. SBSPR2, respectively. Some aspects of the analysis are publicly available on Kbase narrative 92727.

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
