# Peer review of "Methylphosphonate Degradation and Salt-Tolerance Genes of Two Novel Halophilic Marivita Metagenome-Assembled Genomes from Unrestored Solar Salterns"

_genes, 2022, doi:10.3390/genes13010148_

Round 1
Reviewer 1 Report
The paper by Bueno de Mesquita et al. describes the phylogenetic and genetic analysis of two new Marivita strains, based on genomes assembled from metagenomic data. The samples were obtained from unrestored solar salterns in the South San Francisco Bay area. The authors provide a good description of the new species (based on genomic comparisons) and have adequately explained the metagenomic assembly and quality control methods involved. The paper will be valuable in expanding the knowledge of aerobic methane producing bacteria in general and specifically expanding the knowledge of the genus Marivita. Besides a few minor suggestions below, the paper could be accepted in its current form.
Minor suggestions:
- The species/genus and gene names should be italicized in the Result section. It appeared to have been lost during editing.
- Lines 246 and 248: The descriptions of figures 5b and 5c appear to be opposite from the actual figure 5b and c. Figure 5b shows methane flux and Figure 5c shows salinity correlations (while the in-text description is opposite). Please update so the two are aligned.
Author Response
Dear Reviewer,
Thank you for your positive review of our manuscript and for catching the errors in italicization and Figure 5. We have corrected these errors.
Reviewer 2 Report
The manuscript is well written and scientifically sound.
In the results section there are several places where 'Malivita' and the gene names are not written in italics.
Author Response
Dear Reviewer,
Thank you for your positive review and for catching the errors in italicization. We have corrected this. In one case we referred to a protein and have capitalized the first letter but not italicized it.